# Islamic Financial Depth, Financial Intermediation, and Sustainable Economic Growth: ARDL Approach

**Adil Saleem [1], Judit Sági [2,]*** and **Budi Setiawan [1]**

[1] Doctoral School of Economics and Regional Sciences, Hungarian University of Agriculture and Life Sciences, H-2100 Gödöllő, Hungary; saleem.adil@phd.uni-szie.hu (A.S.); setiawan.budi@hallgato.uni-szie.hu (B.S.)

[2] Faculty of Finance and Accountancy, Budapest Business School-University of Applied Sciences, H-1149 Budapest, Hungary

[*] Correspondence: sagi.judit@uni-bge.hu

**Abstract:** The pre-eminence of Islamic finance from the perspective of economic growth has been a long-standing debate. In recent decades, there has been a paradigm shift from interest-based banking to Islamic financial system. This study intends to examine the dynamic interaction of Islamic financial depth (IFD), Islamic financial intermediation (IFI), and asset quality with economic growth in a dual banking system. The paper employs autoregressive distributive lag regression (ARDL), error correction model (ECM) and Granger causality to examine the long and short run linkage by using the quarterly data of Pakistan from 2005 to 2019. The authors run two models to analyze the relative importance of financial depths (Islamic and conventional), financial intermediation (Islamic and conventional), and asset quality of both financial systems. A long-run relationship flowing from finance to growth in both Islamic and conventional finance models has been observed in our study. Furthermore, the findings recommend that strong financial intermediation plays an imperative role in driving economic growth by both financial sectors. The presence of a higher degree of Islamic financial assets in the economy contributes towards economic growth in the short-run. The results show that asset quality possibly plays an important intervening role in the overall finance-growth nexus.

**Keywords:** Islamic banking; conventional banking; asset quality; economic growth; ARDL

**JEL Classification:** O47; G21; E58; F62

## 1. Introduction

The financial sector plays an imperative role in channeling financial resources towards corporate sector where growth opportunities are higher. The debate on financial intermediation and economic growth is rooted back in the theory of Schumpeter (1912), which became the topic of discussion by many economists and researchers. Productive allocation of resources combined with the well-functioning intermediation of financial sector plays a vital part in furthering economic growth (Demirguc-Kunt et al. 2003; King and Levine 1993; Love 2003). Contrary to supply-leading theory, Robinson (1952) and Jung (1986) proposed that a developed economy accelerates financial growth. However, the discussion in the literature led several studies to develop other theories as well, i.e., demand-following, mutual dependence theory, and neutrality phenomenon (Chandler et al. 2017; Lucas 1988; Patrick 1966; Robinson 1952).

The instigation of Islamic finance in the late nineteenth century opened a new paradigm in the financial world. As per the facts highlighted by Ernst and Young (2016), nine core markets constitute 93% percent of international Islamic finance, the major share includes a 33% share by Saudi Arabia, 15.5% contribution from Malaysia, and 15.4% by UAE. In addition, Qatar, Indonesia, Kuwait, Bahrain Pakistan, and Turkey are among the nine core

markets. Furthermore, Muhamad and Mizerski (2013) highlighted the seven largest countries with a dominant Muslim population which includes Egypt, Indonesia, Iran, Nigeria, Pakistan, and Turkey. Pakistan, being the second largest Muslim country with double-digit growth in Islamic banking assets, plays a significant role in Islamizing its banking system. Nevertheless, on the domestic front, an overall 11% of the total banking sector consists of Islamic banking assets that increased to 32.2% of total banking share in Pakistan from 2015 to 2019 (Ernst and Young 2016; SBP 2019a). Furthermore, Pakistan is considered amongst those countries where the Islamic banking sector grew with great potential. The fast-paced growth of the Islamic financial sector in Pakistan started from two branches in 2002, the numbers expanded to five full-fledged Islamic banks with 1563 branches, and 17 conventional banks with 1509 standalone Islamic branches together with 154 sub-branches by the end of 2019 (SBP 2019b). In addition, the contribution to global Islamic banking assets increased from 0.7% in 2010 to 1.7% in 2016 (Ernst and Young 2016). Notably, the central bank (State Bank of Pakistan)—SBP, provides a conducive environment for local and foreign banks to develop and promote Islamic banking. In this regard, in a poll created on Islamic Finance News (IFN)—Malaysia, SBP had won the best central bank award for promoting Islamic banking in 2015 and 2017 (SBP 2020).

Over the last two decades, Pakistan has experienced higher growth in the Islamic financial sector not only on a national level but internationally as well. Therefore, it is highly relevant and of great interest to study the Islamic financial sector and economy of an emerging country like Pakistan. Additionally, it is among the core financial markets that significantly contribute to the global Islamic banking assets. With regard to economic growth, Islamic and conventional finance have been theoretically and empirically compared by many researchers (Szegedi et al. 2020). Several studies have given evidence that Islamic banks turned out to be more stable and efficient (Baber 2018; Hasan and Dridi 2010; Pappas et al. 2017), better in asset quality (Hassan and Hussein 2003), more cost effective (Al-Jarrah and Molyneux 2006), and better in mitigating credit risk (Samad 2004; Samad and Hassan 2006). Although the interaction between Islamic finance and the economy have been studied by many researchers, the results are inconclusive and ambiguous so far. For instance, Chapra (2008) put forward that the presence of Islamic finance allows healthy economic activities, which is an essential element for economic growth. Similarly, according to Khan and Bashar (2008), economic growth is ensured by implementing a developed Islamic banking sector. On the other hand, authors could not find any evidence to support the supremacy of interest-free banking over conventional finance (Furqani and Mulyany 2009).

Studies in favor of Islamic finance and economic growth have been conducted in different countries around the globe. For instance, Jouini (2016) endorsed this relationship in Saudi Arabia, Abduh and Omar (2012), Anwar et al. (2020) in Indonesia, Kassim (2016) in Malaysia, Chowdhury et al. (2018) in Bangladesh, Goaied and Sassi (2010) in the Middle East and North Africa (MENA) region, Lebdaoui and Wild (2016) Southeast Asia, and Abedifar et al. (2016) in 22 Muslim countries. Nevertheless, the findings become inconclusive when other authors explored the insignificance of the relationship between Islamic finance and economic growth in UAE, Bangladesh, Malaysia, Turkey, and MENA countries (Adnan Hye and Islam 2013; Hachicha and Amar 2015; Kar et al. 2011; Yüksel and Canöz 2017; Zarrouk et al. 2017).

Studies estimating the impact of Islamic banking development and economic growth could not provide consistent results in the literature. However, limited attention has been given to this area from the perspective of Pakistan (Asif et al. 2014; Nawaz et al. 2019; Shah and Raza 2020). This study examines the contribution of the Islamic financial system towards economic growth in a dual banking system. The research objective of the study is to examine the relative importance of the Islamic financial system towards economic growth compared to the conventional banking system. Further, our research also determines the insights about the asset quality of financial sector and its impact on economic growth in the long- and short-run.

This paper explores three new areas. Firstly, as the financial sector of Pakistan is governed in a dual banking system, our study focuses on two parallel models examining the impact of Islamic financial depth and conventional financial depth on economic growth. In contrast, previous studies considered overall financial development generally or Islamic financial development. Secondly, many of the studies considered a single indicator of financial development, but our study considers Islamic financial depth and financial intermediation as an indicator of Islamic financial development. The financial intermediation measured by Anwar et al. (2020) is decomposed into two elements: Islamic financial intermediation; and conventional financial intermediation. Thirdly, in previous studies, the element of financial asset quality was completely ignored, which could have a greater impact in influencing the finance–growth nexus.

Our study builds upon existing literature in a number of ways. Firstly, it provides the evidence of a side-by-side comparison of interest based and non-interest based financial depth and its impact on economic growth. Further, the results will help the central bank to discover which financial system (Islamic/conventional) is healthy for economic growth. Secondly, insights of financial intermediation would help central banks and the management of banking companies to analyze the situation of both banking systems. Lastly, understanding the impact of non-performing assets on both financial systems aids in formulating the policies for improving the asset quality of both financial sectors.

By considering the presence of both Islamic and conventional banking within the country's financial intermediary system, our study provides an added value to the understanding of the dual banking systems' effect on economic growth. Hereby, the financial depths, financial intermediation, and asset quality linkages are to be examined by the autoregressive distributed lag (ARDL), error correction model (ECM) and Granger Causality econometric tests.

## 2. Literature Review

The literature review part consists of two sections. The first section will review the links between the Islamic financial sector and economic growth. The connection between financial stability and economic growth is discussed in second part.

### 2.1. Islamic Financial Sector and Economic Growth

A banking system that complies with the Islamic Sharia'h (Islamic legal framework) comes under the definition of Islamic banking. Over the past few decades, the Islamic financial market has experienced remarkable growth around the world. It has proven itself a viable and feasible alternative to a conventional banking system and contributed a larger share to the overall market. Unlike the interest-based banks, Islamic banks work as both a financial intermediary and a trading agent at the same time (Shamsudin et al. 2015). In addition, based on production and intermediation banking approaches, Islamic banks were found to be economically efficient, compared to their traditional counterparts (Musa et al. 2020). Hence, Islamic banks are reactive in promoting real economic activity by participating with their clients as a trading partner, which is essential for economic growth. Literature concerning Islamic finance and economic growth is inconclusive and ambiguous so far and can be divided into four categories.

The first approach of studying the phenomenon of Islamic finance and economic growth is based on the 'Schumpeterian supply-leading' approach. Chowdhury et al. (2018) and Abd. Majid and Kassim (2015) suggested a positive and linear relationship between Islamic banking and economic growth in a Malaysian economic setting. The authors employed bounds testing based on an autoregressive distributed lag (ARDL), variance decomposition, and error correction models to postulate that total Islamic deposits have a positive impact on gross domestic product. However, these authors further argued that financial resource allocation did not appear to have any relationship with growth. Similarly, Hachicha and Amar (2015) investigated the Malaysian economy by considering quarterly data from 2000Q1-2011Q4. The authors used PRIVATE (Islamic bank loans/Total

loan), PRIVIS (Islamic bank loans/GDP), and ENVIS (Islamic bank loans/Investment) as a proxy of Islamic finances to the private sector, Islamic financial depth, and capital accumulation through Islamic finance, respectively. Using Johansen cointegration and Vector error correction model (VECM), the authors found that the long run interaction of Islamic finance is less important, compared to the short run model. In addition, Boukhatem and Moussa (2018) established that Islamic financial depth (finances/GDP) causes economic growth (GDP per capita) in MENA countries in the long run. The study formulated a theoretical framework to establish the connection between Islamic financial development and economic growth. Furthermore, using yearly data from 2000–2014 of 13 countries in the MENA region, the panel cointegration model suggested that a positive relationship between Islamic finance and growth may be affected by a weak institutional framework.

In addition, the supply-leading hypothesis was explored by Lebdaoui and Wild (2016) in Southeast Asia by using a panel ARDL model on quarterly data from 2000 to 2012. The study suggested a strong and significant effect of Islamic financial development on economic growth in the long run. However, authors highlighted a unique connection between large Islamic banks and found a substantial influence on economic progress. In addition, a growing number of studies documented the supportive arguments in favor of the supply-leading theory in different regions globally (Abedifar et al. 2016; Ali and Azmi 2017; Grassa and Gazdar 2014).

The second approach is based on Robinson's demand-following nexus, which states that a developed economy tends to drive financial growth. Many authors studied and documented the causality flowing from growth to the financial sector and stressed that a developed economy could create the demand for investments and savings, which is necessary for financial development. Hassan et al. (2011) examined the financial development and economic growth by employing panel regression and the variance decomposition method on 168 countries' data and stressed a positive causal relationship flowing from growth to finance. Although the authors found a two-way relationship, most of the regions (East Asia, Pacific, Sub-Sahara Africa) appeared to experience the demand-following hypothesis. Similarly, Furqani and Mulyany (2009) explored the Malaysian Islamic financial market and its relationship with economic growth for the period from 1997 to 2005, employing cointegration and VECM. The research documented a two-way causality relationship between Islamic banking and gross fixed capital (GFCF), whereas a demand-following relationship was observed between GDP and Islamic banking in the long run. Furthermore, Zarrouk et al. (2017) in UAE, using time series data from 1990 to 2012, provided a consistent result with Robinson (1952) demand-following theory. Furthermore, the authors argued that the causality flows from GDP to Islamic banking development with no evidence of a reverse direction in UAE.

Regardless of comprehensive literature endorsing the finance-growth linkage Odedokun (1992) argued that finance driven by banks push the economic indicators in a bidirectional relationship. The phenomenon coined by Patrick (1966) presented a mutual dependence hypothesis that states that finance and growth are mutually dependent, and the economic stage of a country would define the actual relationship. Özer and Karagöl (2018) found empirical evidence in Turkey for the mutual dependence hypothesis of fiscal and monetary policy and economic growth. In the context of Islamic finance, Anwar et al. (2020) signified a bidirectional relationship between economic growth, Islamic banking deposits and number of offices. Using ARDL, VECM, VDCs and quarterly data from 2009 to 2019 of the Indonesian Islamic banking industry, the study suggested a significant strong bidirectional (two-way) relationship between Islamic banking development, financial intermediation, and economic growth. Similarly, empirical evidence established a bidirectional relationship between Islamic financial development and economic growth nexus authored by Tabash and Dhankar (2014) when they examined the UAE economy using a cointegration model and annual data from 1990 to 2010. The results suggested that economic growth is led by Islamic financial growth in UAE and vice versa. In addition, relatively coherent outcomes were recorded in Indonesia in 2012 (Abduh and Omar 2012).

Furthermore, Skare and Porada-Rochoń (2019) comprehensively examined the finance-growth nexus in 19 European transition countries using ARDL, ECM and the Granger causality test. The results suggested that in a majority of European transition countries the bi-directional theory persists, whereas the demand-following hypothesis endures in the Macedonian economy. However, a comparatively new addition in theories of the finance-growth nexus is the neutrality hypothesis inferring that economic progress is insusceptible to the financial system of any country and proposed finance-growth as an overstressed dilemma (Lucas 1988). According to Yüksel and Canöz (2017) Islamic finance does not contribute towards progress in the economy of Turkey, neither does the economy support the Islamic banking. The authors postulated that due to a little low market share of Islamic banking in the Turkish financial system, the ability for Islamic finance to drive economic growth ceased to exist. The author used data from 2005 to 2016 and stressed that the Islamic banking sector in Turkey must exist with a larger market share in order to stimulate industrial and economic growth. Similarly, Skare and Porada-Rochoń (2019) found no evidence of the finance-growth nexus in Czech Republic due to the high degree of foreign direct investment (FDI) in the country. However, the large influx of FDI into the country makes the economy less dependent on the financial sector (Sekuloska 2018). In contrast, literature concerning the finance and economic growth nexus also provide a negative relationship in Bangladesh (Adnan Hye and Islam 2013), Saudi Arabia (Samargandi et al. 2014), and Nigeria (Adeniyi et al. 2015).

Research on Islamic finance and growth nexus in the context of Pakistan is limited and has advocated the supply-leading hypothesis so far. For instance, Nawaz et al. (2019), using the unit root test, cointegration, and the granger causality test, explored a positive and significant effect that flows from Islamic finances to economic growth i.e., real GDP per capita. However, authors suggested a bidirectional association between total Islamic financing and gross domestic Investment in the long run. Asif et al. (2014) considered total advances of scheduled banks as a proxy of economic growth, and Islamic bank deposit as a proxy of Islamic banking development.

The study employed unit root and ARDL model to establish the impact of Islamic banking on economic growth using quarterly data from 2002–2012. The study's findings were consistent with Nawaz et al. (2019). Similar to previous studies, Shah and Raza (2020) produced similar results by employing unit root and OLS regression on time series data. Islamic banking financing appeared to be a predictive driver of economic growth, i.e., GDP.

*2.2. Financial Stability and Economic Growth*

Policymakers and economists are now more concerned about the financial stability of the banking sector. The aftermath of the financial crisis led central banks and other regulatory bodies to think about financial stability and its impact on economic progress. Financial instability and exposure to systematic risk of the banking companies is of great importance for the smooth functioning of financial sector (European Central Bank 2017). The debate in empirical literature concerning financial stability and economic growth has two sides, i.e., an instable financial sector would lead to lower economic growth and an external economic shock may lead to financial instability that may complement each other. For instance, Batuo et al. (2018) examined the relationship between financial instability and macroeconomic indicators. They argued that financial instability impedes the flow of the payment system, which leads to deuteriation of economic indicators. Similarly, Duprey et al. (2017) investigated the periods of financial instability using the Markov-switching model and found a negative impact on GDP growth in European countries. Furthermore, similar findings have been documented by many authors in different regions, including Kim and Mehrotra (2017) in Asia and the Pacific region, Misati and Nyamongo (2012) in BRIC countries (i.e., Brazil, Russia, India, China), and Neaime and Gaysset (2018) in MENA countries.

In addition, Creel et al. (2015) examined the relationship between financial instability and economic growth by using the GMM model. The authors used two key measures of

financial stability, i.e., micro-prudential and macro-prudential measures. Micro-prudential stability was measured through non-performing loans to gross loans (following (Čihák and Schaeck 2010; Foglia et al. 2020)), the z-score and IMF stability indicator. The composite indicator of systematic stress (CISS) (following (European Central Bank 2017)) performs as a macro-prudential stability indicator. The results suggested that irrespective of degree of financial depth, financial instability is negatively associated with economic growth. Likewise, Alsamara et al. (2019) investigated the long- and short-run relationship between financial instability and GDP in Qatar. Using the VECM model with structural breaks, the results suggested that GDP is negatively affected by loan provisions in the short and long run. The authors further claimed that the better the asset quality (loan loss provision) of the financial sector, the better the economic progress of a country would be. Manu et al. (2011) considered several measures of financial stability (liquidity, asset quality, capital adequacy) and its impact on economic growth of selected African countries. Using the panel cointegration model, the results revealed that financial instability is negatively associated with economic growth in both the long and short run. In particular, it is importance to consider the financial instability and its impact on economic growth even though there is strong evidence of finance-growth nexus. It may influence the payment structure, negatively influence the consumption, hinder investments, deteriorate real economic activity, and hamper economic growth.

### 2.3. Islamic Finance-Growth Nexus

A review of literature revealed a varied result concerning the Islamic finance–growth nexus. A unanimous outcome could not be established due to the differences in political conditions, economic setting, quality of financial sector, asset quality, efforts contributed by central banks, acceptability of consumers to opt Islamic banks and many other socio-economic factors. Furthermore, some limitations have been identified in previous studies, as most of the studies considered only total Islamic financing as a proxy of Islamic finance development. However, countries contributing a larger share in global Islamic finance market including Turkey, Bangladesh, Malaysia, Indonesia, Saudi Arabia, Qatar, and Pakistan do have a parallel interest-based banking system. In the presence of a dual banking system (i.e., Islamic and conventional), it is highly relevant to study the Islamic finance-growth nexus by considering the conventional banking in line with Islamic financial depth. Secondly, studies conducted in Pakistan (Asif et al. 2014; Nawaz et al. 2019; Shah and Raza 2020) only used total Islamic advances, whereas a relative development of Islamic finance compared to conventional banking should be captured by utilizing the concept of Islamic financial depth (as suggested by Anwar et al. 2020; Elmawazini et al. 2020). Furthermore, the ability of a financial sector (Islamic or conventional) to channel the funds was considered in previous studies, and can be captured by the number of Islamic and conventional branches (Islamic and conventional financial intermediation). A higher number of branches would have better prospects for channeling the funds and hence a greater degree of financial intermediation, provided that internal controls ensure the quality of bank exposures (Lentner et al. 2019). In the end, the asset quality/financial stability plays a significant role in developing a financial sector. To represent the quality of assets, and with this study as the first case to the best of our knowledge, would denote non-performing assets (NPAs) to capture its impact on economic growth. NPA is likely to hinder the financial performance of a financial sector. Most importantly, previously conducted on annual data from Pakistan, one of our significant contributions is to use quarterly time series data using the quadratic match-sum method.

## 3. Data and Research Methods

### 3.1. Data

The study uses time series quarterly data from 2005Q1 to 2019Q4. Data concerning macroeconomic variables is gathered from World Development Indicators (WDI) and International Financial statistics (IMF). Whereas bank specific data is collected from a

Quarterly Performance Review of the Banking sector (SBP) and Quarterly Islamic Banking Bulletins (SBP).

In line with studies by Farahani and Dastan (2013), Hachicha and Amar (2015), Shah and Raza (2020), and Anwar et al. (2020), real GDP is considered as a measure for accessing the overall economic output of Pakistan. GDP is one of the most relevant and principal variables for reflecting the economic progress in any country. Unlike the previous studies, this study uses two separate measures to reflect the relative strength of the financial sector. Therefore, Islamic financial depth is measured through total Islamic banking financing. as a percentage of nominal GDP (Anwar et al. 2020; Boukhatem and Moussa 2018; Elmawazini et al. 2020; Zarrouk et al. 2017), to represent the size of Islamic financial depth in Pakistan. However, conventional financial depth is considered through total loans disbursed by conventional banks as a percentage of nominal GDP. The second important variable for accessing financial intermediation is the number of offices of Islamic banks and conventional banks to represent Islamic financial intermediation and conventional financial intermediation respectively as followed by (Anwar et al. 2020). Our study considers the impact of non-performing assets of both financial sectors on the economic prosperity of the country of interest. Inflation and interest rates are also considered for this study as control variables. We run two separate models that contains six variables, the model (1) includes GDP, IFD (Islamic financial depth), Islamic financial Intermediation (IFI), NPAs (non-performing assets), INF (inflation), and IR (interest rate/KIBOR-Karachi Interbank Offered Rate). Similarly, the second model (2) contains GDP, CFD (conventional financial depth), conventional financial intermediation (CFI), NPLs (non-performing loans), INF (inflation), and IR (interest rate/KIBOR).

$$GDP = f(IFD, IFI, NPAs, INF, IR) \tag{1}$$

$$GDP = f(CFD, CFI, NPLs, INF, IR) \tag{2}$$

With the exception of GDP, all other variables were collected quarterly. However, we employed the quadratic match-sum to convert low frequency GDP to high frequency data. This method is suitable for avoiding seasonal variations and recommended by many authors (Cheng et al. 2012; Faisal et al. 2018). Table 1 provides the data description of the variable used in this study. However, Table A1 (Appendix A) gives a comparison of variables used in existing literature.

### 3.2. Model Specification

Autoregressive distributed lag (ARDL) following Pesaran et al. (2001) is employed to examine the cointegration between Islamic finance and growth. According to Pesaran et al. (2001), ARDL is a least square regression based on an autoregressive with a set of distributed lags for regressors and regressand. In contrast to Johansen and Juselius's (Johansen and Juselius 1990) cointegration method, the ARDL model is more suitable for examining the cointegration between the variables of interest irrespective of degree of integration I(0) or/and I(1) (Haug 2002). Since the sample size of this study is 60, which is a limited number of observations, ARDL is effectively useful for a limited number of observations (Narayan 2005; Pesaran et al. 2001). ARDL estimates give the unbiased results, valid t-statistics, and can control for endogenous regressors (Menegaki 2019). The process of implementing ARDL contains three steps, (i) the first step to check stationarity of data, (ii) the existence of cointegration using a bounds test and (iii) examining the direction of causality (Menegaki 2019). To investigate cointegration among variables of interest, the order of integration is examined using unit root analysis. The Philip Perron (PP) unit root test is performed, which gave us the satisfactory results for using the ARDL model. Since the presence of mixed integrated times series is a precondition for employing ARDL. Table 2 reveals the results obtained from PP unit root tests, which give us a mixed result of variables integrated at level I(0) and at first difference I(1).

**Table 1.** Description of variables.

| Variables | Description | Notation | Data Collected |
|---|---|---|---|
| Economic growth | Real gross domestic product | GDP | World Development Indicators (WDI) |
| Islamic financial depth | Total Islamic financings as a percentage of GDP | IFD | SBP's Islamic Banking Bulletin |
| Islamic financial Intermediation | Represented as number of Islamic banking branches | IFI | SBP's Islamic Banking Bulletin |
| Asset Quality Islamic | Non-performing assets/total finances | NPA | SBP's Islamic Banking Bulletin |
| Conventional financial depth | Total loans as a percentage of GDP | CFD | SBP's Quarterly Banking Review |
| Conventional financial intermediation | Number of conventional banking branches | CFI | SBP's Quarterly Banking Review |
| Asset quality conventional | Non-performing loans/total loans | NPL | SBP's Quarterly Banking Review |
| Inflation | Consumer Price Index | INF | IMF's International Financial Statistics |
| Interest Rate | Discount Rate/KIBOR | IR | IMF's International Financial Statistics |

**Table 2.** Unit Root.

| Variables | Model 1 (Islamic) | | | | Model 2 (Conventional) | | | |
|---|---|---|---|---|---|---|---|---|
| | I(0) | | I(1) | | I(0) | | I(1) | |
| | c | t & c | c | t & c | c | t & c | c | t & c |
| GDP | −0.5818 (0.866) | −1.572 (0.792) | −11.382 * (0.000) | −11.502 * (0.000) | −0.5818 (0.866) | −1.572 (0.792) | −11.382 * (0.000) | −11.502 * (0.000) |
| IFD/CFD | 3.891 (1.000) | 0.242 (0.242) | −7.396 * (0.000) | −9.385 * (0.000) | 0.1819 (0.969) | −1.432 (0.841) | −10.321 * (0.000) | −10.414 * (0.000) |
| IFI/CFI | −5.483 * (0.000) | −10.063 * (0.000) | - | - | −1.366 (0.592) | −3.434 (0.057) | −11.587 * (0.000) | −12.729 * (0.000) |
| NPA/NPL | −1.626 (0.463) | −1.801 (0.691) | −9.727 * (0.000) | −9.799 * (0.000) | −1.237 (0.653) | −1.540 (0.803) | −6.607 * (0.000) | −6.641 * (0.000) |
| INF | 0.982 (0.996) | −1.678 (0.749) | −4.022 * (0.003) | −4.191 * (0.008) | - | - | - | - |
| IR | −1.565 (0.494) | −1.583 (0.788) | −4.337 * (0.001) | −4.333 * (0.006) | - | - | - | - |

* Denotes significance at 1%. Notes: above test is conducted based on Schwarz info criterion with intercept (c) and trend (t).

To estimate the long run cointegration between the variables of interest and GDP, the ARDL bounds test is carried out. Models (1) and (2) can be expressed with the following equation(s).

$$
\begin{aligned}
\Delta \ln(GDP)_t = \ & \pi_0 + \sum_{q=1}^{p1} \vartheta_{1q}\Delta \ln(GDP)_{t-q} + \sum_{q=0}^{p2} \vartheta_{2q}\Delta IFD_{t-q} + \sum_{q=0}^{p3} \vartheta_{3q}\Delta \ln(IFI)_{t-q} \\
& + \sum_{q=0}^{p4} \vartheta_{4q}\Delta NPA_{t-q} + \sum_{q=0}^{p5} \vartheta_{5q}\Delta INF_{t-q} + \sum_{q=0}^{p6} \vartheta_{6q}\Delta IR_{t-q} \\
& + \omega_1 \ln(GDP)_{t-1} + \omega_2 IFD_{t-1} + \omega_3 \ln(IFI)_{t-1} + \omega_4 NPA_{t-1} \\
& + \omega_5 INF_{t-1} + \omega_6 IR_{t-1} + \varepsilon_t
\end{aligned}
\tag{3}
$$

$$
\begin{aligned}
\Delta \ln(GDP)_t = \ & \pi_0 + \sum_{q=1}^{p1} \vartheta_{1q}\Delta \ln(GDP)_{t-q} + \sum_{q=0}^{p2} \vartheta_{2q}\Delta CFD_{t-q} + \sum_{q=0}^{p3} \vartheta_{3q}\Delta \ln(CFI)_{t-q} \\
& + \sum_{q=0}^{p4} \vartheta_{4q}\Delta NPL_{t-q} + \sum_{q=0}^{p5} \vartheta_{5q}\Delta INF_{t-q} + \sum_{q=0}^{p6} \vartheta_{6q}\Delta IR_{t-q} \\
& + \omega_1 \ln(GDP)_{t-1} + \omega_2 CFD_{t-1} + \omega_3 \ln(CFI)_{t-1} + \omega_4 NPL_{t-1} \\
& + \omega_5 INF_{t-1} + \omega_6 IR_{t-1} + \varepsilon_t
\end{aligned}
\tag{4}
$$

Model (1) examines the long- and short-run relationship between Islamic financial depth and the dependent variable, whereas Model (2) is to examine cointegration between conventional financial depth and GDP simultaneously. Where $p_i$ denotes the number of optimal lags used in the ARDL model, $\varepsilon_t$ is the noise, and $\Delta$ denotes the first difference. To access the presence of cointegration ARDL bounds test is carried out to test the null i.e., $H_0 : \omega_1 = \omega_2 = \omega_3 = \omega_4 = \omega_5 = \omega_6 = 0$ against the alternative of presence of cointegration among independent and dependent variables in both Models (1) and (2). According to Pesaran et al. (2001) and Narayan (2005), the computed F-statistics in the bounds test is to be compared with the upper bound and lower bound critical value. If the value of F-statistics in the bounds test is greater than the critical value of upper bound, it could give us reason to reject null of no cointegration in the long run. Furthermore, Narayan (2005) suggested that the ARDL bounds test is even suitable for a limited sample size of 30 to 80 observations.

Once the cointegration is established, a long- run equation can be formed for both the models as shown in Equations (5) and (6).

$$
\begin{aligned}
\ln(GDP)_t = \quad & \pi_0 + \sum_{q=1}^{p1} \gamma_{1q} \ln(GDP)_{t-q} + \sum_{q=0}^{p2} \gamma_{2q} IFD_{t-q} + \sum_{q=0}^{p3} \gamma_{3q} \ln(IFI)_{t-q} \\
& + \sum_{q=0}^{p4} \gamma_{4q} NPA_{t-q} + \sum_{q=0}^{p5} \gamma_{5q} INF_{t-q} + \sum_{q=0}^{p6} \gamma_{6q} IR_{t-q} + \mu_t
\end{aligned}
\tag{5}
$$

$$
\begin{aligned}
\ln(GDP)_t = \quad & \pi_0 + \sum_{q=1}^{p1} \gamma_{1q} \ln(GDP)_{t-q} + \sum_{q=0}^{p2} \gamma_{2q} CFD_{t-q} + \sum_{q=0}^{p3} \gamma_{3q} \ln(CFI)_{t-q} \\
& + \sum_{q=0}^{p4} \gamma_{4q} NPL_{t-q} + \sum_{q=0}^{p5} \gamma_{5q} INF_{t-q} + \sum_{q=0}^{p6} \gamma_{6q} IR_{t-q} + \mu_t
\end{aligned}
\tag{6}
$$

$\mu_t$ refers to the error term in both models. Following Narayan (2005) and Engle and Granger (1987), the error correction model is employed to find the dynamic short-term relationship with variables of interest and the dependent variable. ECM models express the following short-term relationship with the error correction term.

$$
\begin{aligned}
\Delta(\ln(GDP))_t = \quad & \pi_0 + \sum_{q=1}^{p1} \gamma_{1q} \Delta(\ln(GDP))_{t-q} + \sum_{q=0}^{p2} \gamma_{2q} \Delta IFD_{t-q} + \sum_{q=0}^{p3} \gamma_{3q} \Delta(\ln(IFI))_{t-q} \\
& + \sum_{q=0}^{p4} \gamma_{4q} \Delta NPA_{t-q} + \sum_{q=0}^{p5} \gamma_{5q} \Delta INF_{t-q} + \sum_{q=0}^{p6} \gamma_{6q} \Delta IR_{t-q} + \sigma_q ECT_{t-q} + \mu_t
\end{aligned}
\tag{7}
$$

$$
\begin{aligned}
\Delta(\ln(GDP))_t = \quad & \pi_0 + \sum_{q=1}^{p1} \gamma_{1q} \Delta(\ln(GDP))_{t-q} + \sum_{q=0}^{p2} \gamma_{2q} \Delta CFD_{t-q} + \sum_{q=0}^{p3} \gamma_{3q} \Delta(\ln(CFI))_{t-q} \\
& + \sum_{q=0}^{p4} \gamma_{4q} \Delta NPL_{t-q} + \sum_{q=0}^{p5} \gamma_{5q} \Delta INF_{t-q} + \sum_{q=0}^{p6} \gamma_{6q} \Delta IR_{t-q} + \sigma_q ECT_{t-q} + \mu_t
\end{aligned}
\tag{8}
$$

where $\gamma_{1q}$, $\gamma_{2q}$, $\gamma_{3q}$, $\gamma_{4q}$, $\gamma_{5q}$, $\gamma_{6q}$ are the coefficient of short-run equilibrium, $\sigma_i$ is the coefficient of error correction term and represents the movement from disequilibrium for the correction of long run cointegration. ECM and bounds testing only establish cointegration and causality among the variables. Furthermore, we test for the cointegration from both sides of equation by taking each independent variable as dependent variable (the results are given in Tables A2 and A3). However, following Menegaki (2019), Jenkıns and Katırcıoglu (2010) and Geweke et al. (1983) after long-run equilibrium is established using the bounds test, a pairwise granger causality test is performed to examine the direction of causality (the results are given in Tables A2 and A3). The ECM model as represented in Equations (7) and (8) implies the short run dynamic relationship through coefficients of lagged regressors $\gamma_{1q}$, $\gamma_{2q}$, $\gamma_{3q}$, $\gamma_{4q}$, $\gamma_{5q}$, $\gamma_{6q}$ and the long-run equilibrium is captured using error correction term i.e., $ECT_{(t-1)}$. In the ARDL estimation, the selection of optimal lag is of great importance. Before running ARDL, we employed Schwarz Bayesian information criteria (SC) with lag length of $i$, and SC estimated the $(m+1)^4$ lagged

regression models, where m is the maximum lag length selected using SC. Furthermore, the LM Breusch-Godfrey test is conducted to check the insignificance of serial correlation for both models. To check the existence of homoscedasticity, the LM Breusch Pagen test is carried out. On the other hand, to analyze the correct functional form of both models, the Ramsey RESET test is performed. Similarly, normality is satisfied using the Jarque-Berra (JB) normality test.

## 4. Empirical Results and Discussion

In this section, we explain and interpret the outcomes of the statistical model presented in the previous section. First, we provide the base of essential statistical tests that gives us the reason to decide the appropriate cointegration model i.e., ARDL. In addition, Table 3 provides the list of key diagnostic tests for meeting the agreeable assumptions of least square regression.

**Table 3.** Cointegration Test Results.

| Model Specification | F(GDP\|IFD, IFI, NPA) [1] | | F(GDP\|IFD, IFI, NPA, INF, IR) [2] | | F(GDP\|CFD, CFI, NPL) [3] | | F(GDP\|CFD, CFI, NPL, INF, IR) [4] | |
|---|---|---|---|---|---|---|---|---|
| **Bounds Tests** | | | | | | | | |
| F-Statistics | 4.4647 * | | 43.2523 * | | 12.3147 * | | 14.3109 * | |
| Asymptotic Critical Values | K 3 | | K 5 | | K 3 | | K 5 | |
| | I(0) | I(1) | I(0) | I(1) | I(0) | I(1) | I(0) | I(1) |
| 10% | 2.37 | 3.20 | 2.08 | 3.0 | 2.37 | 3.2 | 2.08 | 3.0 |
| 5% | 2.79 | 3.67 | 2.39 | 3.38 | 2.79 | 3.67 | 2.39 | 3.38 |
| 2.5% | 3.15 | 4.08 | 2.70 | 3.73 | 3.15 | 4.08 | 2.70 | 3.73 |
| **Diagnostic tests** | | | | | | | | |
| LM BG $\chi^2$ | 0.751 (0.415) | | 0.402 (0.533) | | 0.761 (0.410) | | 1.096 (0.243) | |
| RESET $\chi^2$ | 1.093 (0.301) | | 0.069 (0.793) | | 2.481 (0.122) | | 1.831 (0.183) | |
| Jarque Bera $\chi^2$ | 1.506 (0.471) | | 0.186 (0.911) | | 4.911 (0.085) | | 0.528 (0.768) | |
| LM BP $\chi^2$ | 3.046 (0.061) | | 1.355 (0.221) | | 1.941 (0.088) | | 0.932 (0.482) | |

* Denotes significance at level 5%. [1] Islamic financial depth (IFD) Model 1(a) autoregressive distributed lag (ARDL) [1,2,1,0] optimal lags are selected based on Schwarz information criteria (SIC). [2] IFD Model 1(b) ARDL [4,3,0,0,2,4] optimal lags are selected based on SIC criteria. [3] conventional financial depth (CFD) Model 2(a) ARDL [1,2,1,0] optimal lags are selected based on SIC criteria. [4] CFD Model 2(b) ARDL [1,3,1,0,1,1] optimal lags are selected based on SIC criteria. Source: authors' calculations using Eviews10.

### 4.1. Bounds Testing

As per Table 1, the time series of both Models (1) and (2) are integrated at level I(0) and/or first difference I(1). According to the non-parametric PP unit root test, the null hypothesis of the presence of unit root has been rejected for all variables at first difference I(1) except IFI, which is stationary at level I(0). To proceed with cointegration estimation, both models are tested for ARDL bounds tests in two ways, i.e., with and without the presence of control variables (inflation and interest rate). However, the models selected seemed to fit and are considered in correct functional form as suggested by the results of diagnostic tests (Table 3). We employed the LM Breusch-Godfrey Test to test for autocorrelation, LM Breusch-Pagen test for Heteroscedasticity, Jarque-Bera test to check the normality, and Ramsey RESET (Regression Equation Specification Error Test) test at level (1) to validate the correctness of functional form of the selected models [1(a), 1(b), 2(a), 2(b)]. The relevant F-stats and Prob $\chi^2$ are included. Results gave us the adequate evidence of normality, homoscedasticity, no autocorrelation, and correct functional form of the model. It also implies that all the models did not deviate from the basic assumption of ordinary least squares (OLS).

Table 3 illustrates the evidence of long-run cointegration of all the models [1(a), 1(b), 2(a), 2(b)]. In the first model of Islamic financial depth (IFD), we ran two models to

validate the strong evidence of long-term cointegration between Islamic financial depth and economic growth. In the first place, we exclude the control variable i.e., inflation and interest rate and then we included all five predicting variables to access the long-run integration of Islamic finance and GDP. In a similar fashion, we did the same for model of conventional financial depth (CFD) to assess how strongly conventional financial depth has been integrated with GDP in the long run. The asymptotic F-Statistics for the bounds tests are not standard; hence we compared the F-statistics of ARDL bounds tests with lower and upper bound critical values. If the values of F-stats are above the upper critical value I(1), we can infer the presence of cointegration of the explanatory variables and dependent variable in the ARDL model (Pesaran et al. 2001). On the other hand, if the value falls below the lower critical value I(0), it depicts evidence of no cointegration. There is another possibility of inconclusive cointegration: if the calculated F-statistics fall between I(0) and I(1). For Islamic financial depth, we run two ARDL bounds tests for Model-1(a) F(GDP | IFD, IFI, NPA), and 1(b) F(GDP | IFD, IFI, NPA, INF, IR). For Model 1(a) and 1(b), the F-statistics are 4.464 and 43.252 respectively, which is clearly greater than upper critical value I(1) at 5% significance i.e., 3.67 and 3.38. The variables Islamic financial depth, Islamic financial intermediation and Non-performing assets are integrated in the long run with the economic growth of the country. Hence, the same is also true in the presence of control variables at 5% level of significance. Parallel to Islamic financial depth, our model measures the level of integration of conventional financial depth and economic growth as decomposed into two models. The first model, 2(a), represented as F(GDP | CFD, CFI, NPL) while the second model—2(b) is illustrated as F(GDP | CFD, CFI, NPL, INF, IR). F-statistics calculated through ARDL bounds test are 12.314 and 14.310 for Model 2(a) and 2(b) respectively. The results show the upper bound critical values for both models are 3.67 and 3.38 respectively, which are undoubtedly less than the calculated F values. Hence, the null hypothesis of the bounds test, i.e., no cointegration is rejected both all the models. There is strong evidence of long-run integration of both Islamic financial depth and conventional financial depth in predicting the economic growth in Pakistan.

### 4.2. Long Run Elasticities and Granger Causality

In the second stage, as the cointegration is established, we estimate the long run association using ARDL long-run estimate which uses Equations (5) and (6) specified in the previous section. The long run estimates of first Islamic financial depth Model 1(a) (GDP | IFD, IFI, NPA) is based on ARDL [1,2,1,0], optimal lags, selected on the basis on SIC criteria. Table 4 illustrates the results of the long-run relationship of Islamic financial depth, intermediation, and asset quality with economic growth. A long-run relationship is observed between IFD and economic growth, which is significant at 5%. The coefficient of IFD is 0.0162, which implies that if the degree of Islamic financial depth increases by one percent it would cause a 1.6% increase in economic growth. Furthermore, the Granger causality test suggests a significant unidirectional relationship from IFD to GDP (see Table A3). Our result related to IFD is in line with previous literature (Boukhatem and Moussa 2018; Hachicha and Amar 2015; Kassim 2016) Similarly, the ability of Islamic financial institutions to foster intermediation is measured through the growing number of branches. Consistent with (Anwar et al. 2020), the Islamic financial intermediation is found to have a positive significant long-run relationship with economic growth and also have a significant Granger causal relationship from GDP to IFI. The results imply that a one percent change in the measure of Islamic financial intermediation is expected to cause almost a 6% change in the economic growth of the country. However, the coefficient of NPA is negative but not significant, and this implies that NPA hinders growth, but their effect is not statistically significant. The negative relationship is consistent with the previous finding of Alsamara et al. (2019). In a similar fashion, our main predicting model for long-run integration is represented as 1(b) F(GDP | IFD, IFI, NPA, INF, IR), and long run ARDL [4,3,0,0,2,4] is estimated based on SIC lag selection criteria. The results indicate that in presence of inflation and interest rate as control variables, the predictability power of Islamic financial depth and Islamic

financial intermediation is positive and significant in the long run. The coefficients of IFD and IFI are 0.0213 and 0.0358, which are significant at 1%. NPA and IR are negatively associated with GDP in the long run, but IR is found to have a significant relationship with economic growth. Whereas inflation appeared to be reactive for economic growth in the long run with a coefficient of 0.0014 at 1% level of significance. Although inflation's impact on GDP is weak, it has a positive and statistically significant long-run relationship with GDP. The positive impact of inflation on economic growth is contradictory to previous studies (Boukhatem and Moussa 2018; Goaied and Sassi 2010). However, in view of Keynesian, and neo-Keynesian theories our findings are consistent with the results of Saleem and Ashfaque (2020), which states that well predicted inflation has a positive impact on financial performance. Predicted inflation would have a positive impact on economic growth in the long run.

**Table 4.** Long run estimates.

| Model Specification | F(GDP\|IFD, IFI, NPA) [1] | F(GDP\|IFD, IFI, NPA, INF, IR) [2] | F(GDP\|CFD, CFI, NPL) [3] | F(GDP\|CFD, CFI, NPL, INF, IR) [4] |
|---|---|---|---|---|
| IFD/CFD | 0.0162 (2.522) ** | 0.0213 (16.101) *** | −0.0015 (−0.588) | 0.0093 (2.403) ** |
| IFI/CFI | 0.0579 (2.870) *** | 0.0358 (5.097) *** | 1.141 (10.999) *** | 0.672 (2.179) ** |
| NPA/NPL | −0.014 (0.176) | −0.0003 (−0.140) | −0.0095 (−2.975) *** | −0.0023 (−0.425) |
| INF | - | 0.0014 (6.430) *** | - | 0.0001 (0.098) |
| IR | - | −0.0097 (−3.752) *** | - | −0.0078 (−1.949) * |
| - | 21.247 (154.62) *** | 21.318 (371.55) *** | 11.352 (13.262) *** | 15.334 (5.741) *** |

*,**,*** Denotes significance at level 10%, 5%, 1% respectively. [1] IFD Model 1(a) ARDL [1,2,1,0] optimal lags are selected based on SIC criteria. [2] IFD Model 1(b) ARDL [4,3,0,0,2,4] optimal lags are selected based on SIC criteria. [3] CFD Model 2(a) ARDL [1,2,1,0] optimal lags are selected based on SIC criteria. [4] CFD Model 2(b) ARDL [1,3,1,0,1,1] optimal lags are selected based on SIC criteria. Source: authors' calculations using Eviews10.

The long-run relationship between conventional financial depth and economic growth is analyzed with two long-run ARDL models F(GDP|CFD, CFI, NPL) and F(GDP|CFD, CFI, NPL, INF, IR) with optimal lags ARDL [1,2,1,0] and ARDL [1,3,1,0,1,1] selected through SIC criteria. The long run dynamics of conventional financial depth is negative and insignificant, which indicates that in the long-run loans extended by conventional banks do not contribute to the economic growth. However, there is a strong and significant long-term relationship found between financial intermediation and economic growth. The coefficient of CFI is 1.141, which is significant at 1%. However, in the presence of control variables, CFD was found to have a positive and significant long-run relationship with economic growth, although the coefficient is weaker than the Islamic financial depth, and both are significant at 5%. Financial intermediation has a long-term positive and significant contribution toward economic growth. The coefficient of CFI is 0.672 which is significant at 1%. Like Islamic financial depth, the interest rate is significantly and negatively related to economic growth in the long run having a weak coefficient of −0.0078 at 10% level of statistical significance. Inflation and NPLs do not have significant long run relationships. The contribution of Islamic financial depth and conventional financial depth in both the models are significant and positive in the long run. However, the coefficient of IFD is much higher than CFD suggesting that a one percent change in Islamic financial depth would have a greater impact on economic growth in the long run.

### 4.3. Short Run and ECM

Now we estimate the short-term dynamic using the ECM form. Error correction models tells us the short run association between explanatory variables and dependent variable. This model enables us to draw the conclusion about the speed adjustment from the short run disequilibrium to long run equilibrium. In both models F(GDP|IFD, IFI,

NPA, INF, IR) and F(GDP | CFD, CFI, NPL, INF, IR) the coefficient of ECT(-1) is negative and significant 5%.

The negative coefficients suggest the speed of adjustment to attain the long run equilibrium. In our first model, almost 80% of the movements from disequilibrium in the short run tends to adjust in the next quarter to converge into long run integration. Whereas the conventional financial depth model found a slower speed of adjustment with a coefficient of $-0.423$. As per the equilibrium term, the Islamic financial sector appeared to converge towards equilibrium with a greater speed of adjustment. However, the speed of convergence for traditional financial depth is 50% slower than the Islamic depth. Table 5 provides the results of short run association between the variables of interest. In the short run, Islamic financial depth (0.0131), inflation ($-0.0029$) and interest rate (0.0053) have a short-term relationship with economic growth, and coefficients are significant at 5%. This implies that Islamic financial activities contribute towards attaining economic growth in the short run. However, the GDP elasticity in the short run for conventional financial depth and intermediation is weak and not statistically significant. In addition, the coefficient of short run impact of Islamic finance is weak compared to the long run coefficient. Hence, in terms of sensitivity of GDP elasticity, our results are inconsistent (Hachicha and Amar 2015). On the other hand, our short run estimates are consistent with the previous economic finding of Kassim (2016) in Malaysia, Zarrouk et al. (2017) in UAE, Yusof and Bahlous (2013) in GCC, and Abduh and Omar (2012) in Indonesia. Only the interest rate is associated with economic growth in CFD model. Furthermore, the asset quality of Islamic banks and conventional banks are negatively associated with economic growth (Alsamara et al. 2019; Creel et al. 2015), though the results are not significant.

### 4.4. Robustness and Reliability

To test the robustness of the model employed, we applied the LM Breusch-Godfrey Test to test for autocorrelation, LM Breusch-Pagen test for Heteroscedasticity, Jarque-Bera test to check the normality, and Ramsey RESET test at level (1) to validate the correctness of functional form of the selected models [1(a), 1(b), 2(a), 2(b)]. The relevant F-stats and Prob $\chi^2$ are included. Hence, Prob $\chi^2 > 5\%$, showing all estimates are reliable. Therefore, all diagnostic tests gave adequate evidence of normality, homoscedasticity, no autocorrelation, and correct functional form of the model.

Furthermore, model stability is measured using cumulative sum of recursive residuals (CUSUM) as suggested by Pesaran et al. (2001) is conducted to validate the stability of both models (Equations (3) and (4)). Furthermore, Table 3 shows the other diagnostic tests and gave us the satisfactory results to make our model fit for analysis. Figure 1 shows the representation of CUSUM tests for Equations (3) and (4), which is significant at 5%.

**Table 5.** Short run estimates using ECM.

| Differenced Variables | F(GDP\|IFD, IFI, NPA, INF, IR) [1] | | F(GDP\|CFD,CFI,NPL,INF, IR) [2] |
| --- | --- | --- | --- |
| D(GDP(-1) | −0.0103 (−0.0646) | D(GDP(-1)) | 0.0788 (0.6769) |
| D(GDP(-2)) | −0.1336 (−1.1915) | D(GDP(-2)) | - |
| D(IFD) | 0.0131 (2.7721) * | D(CFD) | 0.00045 (0.3123) |
| D(IFD(-1)) | 0.00328 (0.7914) | D(CFD(-1)) | 0.00214 (1.4799) |
| D(IFI) | −0.0107 (0.8587) | D(CFI) | −0.6150 (0.0017) |
| D(IFI(-1)) | - | D(CFI(-1)) | - |
| D(NPA) | −0.0018 (−0.8362) | D(NPL) | 0.00121 (0.3178) |
| D(NPA(-1)) | - | D(NPL(-1)) | - |
| D(INF) | −0.0029 (−3.500) * | D(INF) | −0.0038 (−3.7602) |
| D(INF(-1)) | −0.0028 (−2.9840) * | D(INF(-1)) | - |
| D(IR) | −0.00111 (−0.5919) | D(IR) | 0.00305 (1.1787) |
| D(IR(-1)) | 0.00528 (2.5690) * | D(IR(-1)) | - |
| C | 0.0272 (6.0540) * | | 0.02352 (5.6808) * |
| ECT(-1) | −0.7987 (−4.5593) * | ECT(-1) | −0.4236 (−3.4756) * |
| Adjusted R[2] | 0.8615 | | 0.6800 |
| Log likelihood | 208.7563 | | 184.8972 |
| F-statistic (prob = 0.00) | 23.8144 | | 14.2236 |
| AIC [3] | −6.8842 | | −6.1367 |
| SIC [3] | −6.3055 | | −5.7783 |

* Denotes significance at level 5%. [1] IFD Model 1(b) ARDL [4,3,0,0,2,4] optimal lags are selected based on SIC criteria. [2] CFD Model 2(b) ARDL [1,3,1,0,1,1] optimal lags are selected based on SIC criteria. [3] Akaike info criteria, Schwarz info criteria.

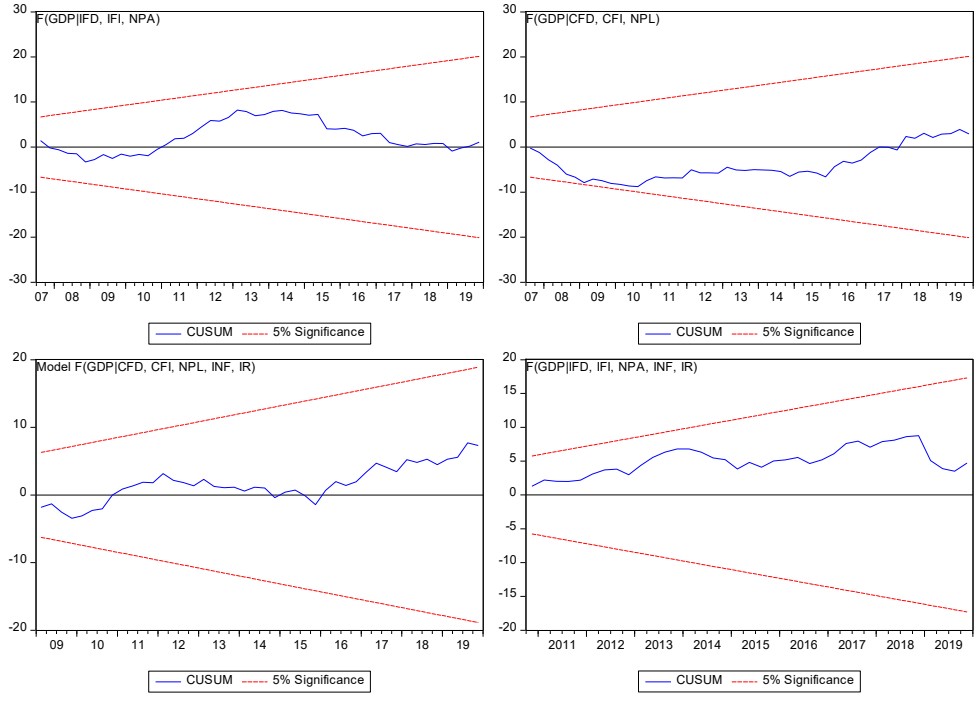

Source: Authors calculations using Eview10

**Figure 1.** CUSUM test.

## 5. Conclusions

The prime objective of this study was to estimate the impact of the Islamic financial sector on economic growth in the presence of a conventional banking system. We developed two models to access how Islamic financial depth, conventional financial depth, and economic growth are connected in the long and short run. We employed ARDL and ECM models on quarterly data of Pakistan financial sector from 2005Q1 to 2019Q4. The unit root test confirmed the presence of stationarity at the first order. The ARDL bounds test was conducted on the selected models to establish the evidence of cointegration. The Granger causality test was performed to analyze the direction of causality in the long run.

The results suggest that the deepness of the Islamic financial sector in the economy contributes equally compared to traditional finance towards achieving economic growth in a country where a dual banking system is prevalent. However, our finding implies that the eminence of the conventional banking system in facilitating real economic activity and fostering economic growth is no less than Islamic finance. In addition, Granger causality tests suggest that both IFD and CFD cause GDP to grow in the long run. Although IFD and CFD significantly affect economic growth in the long run, the strength of the coefficient of Islamic financial depth is much larger than CFD. The linkage of IFD and GDP become stronger given that the growth of Islamic finance is exponentially higher than the interest-based finance. Importantly, the competence of Islamic finance in bringing sustainable economic growth is largely dependent on how firmly it is implemented in a country.

We discovered a unique linkage between financial intermediation and GDP growth. IFI and CFI are both associated with GDP in the long run, but the direction of causality presented interesting information. Islamic financial intermediation is greatly influenced by economic growth and causality flows from GDP to IFI. However, a bidirectional relationship has been observed for CFI and GDP. Our findings suggest that the traditional banking system is working at maturity and captures a larger share of financial sector. Hence, financial intermediation originating from interest-based banks causes GDP to grow and vice versa. Moreover, asset quality could play a significant role in both banking systems and their relative impact on economic growth. We found a negative relationship between NPA, NPL, and GDP. Although the long run impact of NPA and NPL on GDP is not significant, it hinders the financial sector in contributing to economic activity. Islamic financial depth significantly affects economic growth in the short run whereas the same is not true for the conventional banking system. However, in interest-based financing, the benefit could not be achieved in the short run. Similarly, conventional financial intermediation is negatively associated with GDP in the short run whereas no evidence was found for a short run relationship of IFI and GDP.

Based on the statistical evidence, our study implies that in countries with dual banking systems, central banks must strive to implement and develop an Islamic financial system. As highlighted in the results, economic growth is more sensitive to Islamic financial depth compared to the conventional banking sector in both the long and short run. Central banks need to formulate policies to encourage investors, manufacturers and other businesses to promote the Islamic financial system. This may encourage capital accumulation, the manufacturing process, and ultimately encourage suitable economic conditions.

The second important implication of our research is for banks' management to focus on enhancing the branch network to increase financial intermediation. Islamic banks should be operative parallel to conventional counterparts so that final consumers have an opportunity to choose the bank of their choice. Furthermore, asset quality should be given importance by the management as it might hinder the process of economic output in the long run.

There are some areas through which research can be refined in the domain of the Islamic finance–growth nexus. Countries with dual banking systems should be considered by examining both systems at the same time. Future research should consider the element of heterogeneity among countries with dual banking systems and size of financial depth. For Islamic finance, the financing side of the balance sheet may be decomposed on the type of financing and the type of finance that triggers the economic sector may be examined. Furthermore, the mediating role of asset quality (i.e., NPA, NPL) might be examined through its interaction with total financing.

**Author Contributions:** A.S. and J.S. worked on the conceptualization and research design, investigation, data analysis, and methodology. B.S. have contributed to methodology, software, and discussion part. However, J.S. also contributed in reviewed, supervised, and editing of the paper. All authors have read and agreed to the published version of the manuscript.

**Funding:** This research received no external funding.

**Institutional Review Board Statement:** Not applicable.

**Informed Consent Statement:** Not applicable.

**Data Availability Statement:** Publicly available datasets were analyzed in this study. This data can be found from WDI, IMF, and SBP.

**Acknowledgments:** The authors would like to thank to the Hungarian University of Agriculture and Life Sciences, Budapest Business School, and Tempus Public Foundation for their support. This may include administrative and technical support, or donations in kind (e.g., materials used for experiments).

**Conflicts of Interest:** The authors declare no conflict of interest.

**Appendix A**

**Table A1.** Description of variables using the literature.

| Authors | Dependent Variable | Independent Variable | Model |
|---|---|---|---|
| Hachicha and Amar (2015) | Real GDP | Private = Islamic bank loans/total loan.<br>PRIVIS = Islamic bank loans/GDP.<br>ENVIS = Islamic bank loans/investment.<br>Capital = GFCF.<br>Labor force. | Engle and Granger Cointegration model (Engle and Granger 1987). |
| Shah and Raza (2020) | Real GDP | Total Islamic banking financing.<br>Broad money.<br>GFCF.<br>Labor force.<br>Trade openness. | Ordinary least square (OLS) method, and Granger causality test |
| Anwar et al. (2020) | Real GDP | Total deposit/GDP<br>Total financing/GDP<br>No of branches | ARDL [1], VECM [2], VDCs [3], IRFs [4] |
| Boukhatem and Moussa (2018) | Real GDP | Islamic banking development = loans/GDP.<br>GDP per capita.<br>Inflation.<br>Education and human index.<br>Government dxpenditure.<br>Trade openness.<br>Domestic loans to private sector/GDP.<br>Rule of law.<br>Regulatory quality. | Pedroni and Westerlund Panel Cointegration |
| Elmawazini et al. (2020) | GDP Per capita | Islamic banks financing/GDP.<br>Conventional bank financing/GDP.<br>Total financing/GDP.<br>Interaction of rule of law. | Cross-sectionally Correlated and timewise autoregressive (CCTA) model |
| Zarrouk et al. (2017) | GDP Per capita | Financial depth = M2/GDP<br>Financial intermediation = credit by financial sector/GDP<br>Banking development = credit by banks/GDP<br>Islamic banking development = Islamic financial investment/GDP. | Bivariate vector autoregressive model |
| Farahani and Dastan (2013) | Real GDP | Islamic financing<br>GFCF<br>Trade openness | ARDL [1], ECM [5], VAR [6] |
| Kassim (2016) | Industrial production index | Total deposit of Islamic banks<br>Total financing of Islamic banks<br>GFCF<br>Government expenditure<br>Trade openness<br>Inflation | Engle and Granger cointegration model (Engle and Granger 1987). |
| Abduh and Omar (2012) | Ln of GDP | Ln of Deposit<br>Ln of Financing/loans | ARDL [1], ECM [5] |
| Asif et al. (2014) | Real GDP | Advances<br>Inflation<br>Interest rate<br>FDI | ARDL [1] |
| Abedifar et al. (2016) | GDP Per capita | Deposit/GDP<br>Total deposit in the financial system<br>Private credit/GDP<br>Inequality and poverty = GINI index<br>Islamic banking share | Panel Regression |
| Yusof and Bahlous (2013) | GDP Per capita | Loan/GDP<br>Stock market development = stock/GDP<br>GFCF/GDP<br>Trade openness | Pedroni Panel Cointegration VDCs [3] IRFs [4] |
| Furqani and Mulyany (2009) | GDP, GFCF, TRADE | Islamic banks total financing | VECM [2] |

[1] Autoregressive distributive lag. [2] Vector error correction model. [3] Variance decomposition. [4] Impulse response function. [5] Error correction model. [6] Vector autoregressive.

**Table A2.** Cointegration test results.

| Model Specification | F(IFD \| GDP, IFI, NPA, INF, IR) [1] | F(IFI \| GDP, IFD, NPA, INF, IR) [2] | F(NPA \| GDP, IFD, IFI, INF, IR) [3] | F(INF \| GDP, IFD, IFI, NPA, IR) [4] | F(IR \| GDP, IFD, IFI, NPA, INF) [5] |
|---|---|---|---|---|---|
| **Bounds Tests** | | | | | |
| F-Statistics | 4.619 * | 7.477 * | 4.593 * | 2.393 | 6.751 * |
| Asymptotic Value | K-5 | K-5 | K-5 | K-5 | K-5 |
| | I(0) | I(1) | I(0) | I(1) | I(0) | I(1) |
| | 3.41 | 4.68 | 3.41 | 4.68 | 3.41 | 4.68 |
| | 2.62 | 3.79 | 2.62 | 3.79 | 2.62 | 3.79 |
| | 2.26 | 3.35 | 2.26 | 3.35 | 2.26 | 3.35 |
| **Diagnostic tests** | | | | | |
| LM BG $\chi^2$ | 0.932 | 0.060 | 0.558 | 0.245 | 0.188 |
| Ramsey RESET $\chi^2$ | 0.537 | 0.081 | 0.827 | 0.068 | 0.658 |
| JB $\chi^2$ | 0.527 | 0.045 | 0.163 | 0.993 | 0.182 |
| LM BP $\chi^2$ | 0.777 | 0.107 | 0.436 | 0.374 | 0.718 |
| **Pairwise Granger causality test** | | | | | |
| | | F statistics | Prob | Direction | |
| IFD→ | GDP | 17.025 | 0.000 | IFD → GDP | |
| GDP→ | IFD | 2.379 | 0.102 | | |
| IFI→ | GDP | 1.829 | 0.170 | GDP → IFI | |
| GDP→ | IFI | 3.607 | 0.034 | | |
| NPA→ | GDP | 3.260 | 0.046 | NPA → GDP | |
| GDP→ | NPA | 0.208 | 0.813 | | |
| INF→ | GDP | 5.970 | 0.004 | INF ⇌ GDP | |
| GDP→ | INF | 7.768 | 0.001 | | |
| IR→ | GDP | 1.599 | 0.212 | IR — GDP | |
| GDP→ | IR | 0.240 | 0.787 | | |
| IFI→ | IFD | 0.484 | 0.619 | IFI — IFD | |
| IFD → | IFI | 0.907 | 0.410 | | |
| NPA→ | IFD | 0.778 | 0.464 | NPA — IFD | |
| IFD→ | NPA | 0.307 | 0.737 | | |
| INF→ | IFD | 2.076 | 0.136 | IFD → INF | |
| IFD→ | INF | 10.759 | 0.000 | | |
| IR→ | IFD | 6.461 | 0.003 | IR → IFD | |
| IFD→ | IR | 0.629 | 0.537 | | |
| NPA→ | IFI | 1.642 | 0.203 | NPA — IFI | |
| IFI→ | NPA | 1.067 | 0.351 | | |
| INF→ | IFI | 1.134 | 0.329 | IFI — INF | |
| IFI→ | INF | 3.194 | 0.049 | | |
| IR→ | IFI | 1.545 | 0.223 | IR — INF | |
| IFI→ | IR | 0.403 | 0.671 | | |
| INF→ | NPA | 1.071 | 0.349 | INF — NPA | |
| NPA→ | INF | 1.046 | 0.359 | | |
| IR→ | NPA | 8.709 | 0.001 | IR → NPA | |
| NPA→ | IR | 0.188 | 0.829 | | |
| IR→ | INF | 2.748 | 0.073 | INF → IR | |
| INF→ | IR | 3.338 | 0.043 | | |

* Denotes significance at level 5%. [1] IFD Model ARDL [1,4,2,2,1,0] optimal lags are selected based on SIC criteria. [2] IFI Model ARDL [1,2,3,0,4,4] optimal lags are selected based on SIC criteria. [3] NPA Model ARDL [2,2,0,0,1,1] optimal lags are selected based on SIC criteria. [4] INF Model ARDL [1,0,0,4,3,4] optimal lags are selected based on SIC criteria. [5] IR Model ARDL [2,3,3,0,1,3] optimal lags are selected based on SIC criteria. Note: '→', '⇌', and '—' represent the unidirectional, bidirectional, and no evidence of causality, respectively. Source: authors' calculations using Eviews10.

**Table A3.** Cointegration test results.

| Model Specification | F(CFD\|GDP, CFI, NPA, INF, IR) [1] | F(CFI\|GDP, CFD, NPA, INF, IR) [2] | F(NPL\|GDP, CFD, CFI, INF, IR) [3] | F(INF\|GDP, CFD, CFI, NPA, IR) [4] | F(IR\|GDP, CFD, CFI, NPA, INF) [5] |
|---|---|---|---|---|---|
| | **Bound Tests** | | | | |
| F-Statistics | 3.289 * | 2.249 | 5.369 * | 2.357 | 6.792 * |
| Asymptotic Value | k-5 | k-5 | k-5 | k-5 | k-5 |
| I(0) | I(1) | I(0) | I(1) | I(0) | I(1) |
| 3.41 | 4.68 | 3.41 | 4.68 | 3.41 | 4.68 |
| 2.62 | 3.79 | 2.62 | 3.79 | 2.62 | 3.79 |
| 2.26 | 3.35 | 2.26 | 3.35 | 2.26 | 3.35 |
| | **Diagnostic tests** | | | | |
| LM BG $\chi^2$ | 0.513 | 0.163 | 0.989 | 0.193 | 0.352 |
| Ramsey RESET $\chi^2$ | 0.536 | 0.060 | 0.141 | 0.991 | 0.859 |
| JB $\chi^2$ | 0.829 | 0.804 | 0.356 | 0.806 | 0.868 |
| LM BP $\chi^2$ | 0.097 | 0.120 | 0.385 | 0.974 | 0.139 |

| | | **Pairwise Granger causality test** | | |
|---|---|---|---|---|
| | | F statistics | Prob | Direction |
| CFD→ | GDP | 4.052 | 0.023 | CFD → GDP |
| GDP→ | CFD | 2.733 | 0.074 | |
| CFI→ | GDP | 6.103 | 0.004 | GDP ⇌ CFI |
| GDP→ | CFI | 10.484 | 0.000 | |
| NPL→ | GDP | 1.516 | 0.229 | NPL — GDP |
| GDP→ | NPL | 1.582 | 0.215 | |
| INF→ | GDP | 5.970 | 0.005 | INF ⇌ GDP |
| GDP→ | INF | 7.768 | 0.001 | |
| IR→ | GDP | 1.599 | 0.212 | IR — GDP |
| GDP→ | IR | 0.240 | 0.787 | |
| CFI → | CFD | 0.528 | 0.593 | CFI — CFD |
| CFD→ | CFI | 1.646 | 0.203 | |
| NPL→ | CFD | 3.586 | 0.035 | NPL → CFD |
| CFD→ | NPL | 2.592 | 0.084 | |
| INF → | CFD | 0.161 | 0.852 | INF → CFD |
| CFD→ | INF | 8.239 | 0.001 | |
| IR→ | CFD | 3.640 | 0.033 | IR → CFD |
| CFD→ | IR | 0.433 | 0.651 | |
| NPL→ | CFI | 0.272 | 0.763 | NPL — CFI |
| CFI→ | NPL | 1.001 | 0.375 | |
| INF→ | CFI | 3.405 | 0.041 | INF → CFI |
| CFI→ | INF | 2.661 | 0.079 | |
| IR→ | INF | 0.648 | 0.527 | IR — INF |
| INF→ | IR | 1.026 | 0.366 | |
| INF→ | NPL | 1.537 | 0.225 | INF — NPL |
| NPL→ | INF | 0.381 | 0.685 | |
| IR→ | NPL | 7.256 | 0.001 | IR → NPL |
| NPL→ | IR | 1.758 | 0.182 | |
| IR→ | INF | 2.748 | 0.073 | INF → IR |
| INF→ | IR | 3.338 | 0.043 | |

* Denotes significance at level 5%. [1] CFD Model ARDL [3,4,0,0,4,2] optimal lags are selected based on SIC criteria. [2] CFI Model ARDL [5,2,2,0,1,0] optimal lags are selected based on SIC criteria. [3] NPL Model ARDL [2,0,0,0,2,1] optimal lags are selected based on SIC criteria. [4] INF Model ARDL [1,4,1,0,4,3] optimal lags are selected based on SIC criteria. [5] IR Model ARDL [1,1,0,0,1,1] optimal lags are selected based on SIC criteria. Note: '→', '⇌', and '—' represent the unidirectional, bidirectional, and no evidence of causality, respectively. Source: authors' calculations using Eviews10.

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
