# Peer review of "Islamic Financial Depth, Financial Intermediation, and Sustainable Economic Growth: ARDL Approach"

_economies, doi:10.3390/economies9020049_

Round 1
Reviewer 1 Report
Referee report
Islamic financial depth, Financial intermediation, and sustainable Economic Growth: ARDL Approach
The paper investigates the potential positive impact of Islamic financial sector on economic growth and economic growth sustainability. The paper gather data from international institutions to test those relationships within a panel of countries by using a well-established econometric approach. Even if no strong methodological improvements are supplied, the paper offers an interesting review, an interesting application with an econometric exercise for Pakistan, and it develops solid policy insight that can add on the debate. For this reason, I would encourage acceptance after a set of issues are carefully addressed:
- I think that to enable major relevance, the value added has to be emphasized in the latest part of the Introduction. How practitioners can benefit from the econometric test? What is the value added of the insights? What can economy learn from the presence of Islamic finance? All these features are in the paper, but everything should be enclosed in a clear-cut within the Introduction.
- In the Review part (2) a large part is devoted to explain the functioning and potential benefits of Islamic financial system. What it would be really interesting is to enlarge this discussion to the theme of financial stability. In fact, over the last years a great debate has involved the role of financial stability in the economic system with consequences on economic growth (among others, Creel et al. 2015; Foglia et al. 2020).
- The use of the ARDL methodology is interesting, but discussed too briefly. I would add a bit on the technique as in Menegaki (2019) so that it would help to follow the way the testing procedures as well as the assumptions made. This, in order to guarantee results to be more robust.
- I would add more on the role of the Islamic finance in short run consequences. A discussion part could be added. Additionally, comparisons with previous studies should be embedded (e.g., Kassim, 2016).
References
Creel, J., Hubert, P., & Labondance, F. (2015). Financial stability and economic performance. Economic Modelling, 48, 25-40.
Foglia, M., Cartone, A., & Fiorelli, C. (2020). Structural differences in the Eurozone: Measuring financial stability by Fci. Macroeconomic Dynamics, 24(1), 69-92.
Kassim, S. (2016). Islamic finance and economic growth: The Malaysian experience. Global Finance Journal, 30, 66-76.
Menegaki, A. N. (2019). The ARDL method in the energy-growth nexus field; best implementation strategies. Economies, 7(4), 105.
Author Response
Response to Reviewer 1 Comments
Point 1: I think that to enable major relevance, the value added has to be emphasized in the latest part of the Introduction. How practitioners can benefit from the econometric test? What is the value added of the insights? What can economy learn from the presence of Islamic finance? All these features are in the paper, but everything should be enclosed in a clear-cut within the Introduction.
Response 1: Agree with the reviewer. We enclosed these issues in lines 97-99, 101, 112, 121-125
Point 2: In the Review part (2) a large part is devoted to explain the functioning and potential benefits of Islamic financial system. What it would be really interesting is to enlarge this discussion to the theme of financial stability. In fact, over the last years a great debate has involved the role of financial stability in the economic system with consequences on economic growth (among others, Creel et al. 2015; Foglia et al. 2020).
Response 2: Agree. We wrote about financial stability in the Literature Review part, in lines 128-130, 240-279.
Point 3: The use of the ARDL methodology is interesting, but discussed too briefly. I would add a bit on the technique as in Menegaki (2019) so that it would help to follow the way the testing procedures as well as the assumptions made. This, in order to guarantee results to be more robust.
Response 3: Agree. We extended our model specification in lines 346-356.
Point 4:
I would add more on the role of the Islamic finance in short run consequences. A discussion part could be added. Additionally, comparisons with previous studies should be embedded (e.g., Kassim, 2016).
Response 4: Agree. We added more about the role of Islamic finance, and some parts as comparisons with previous studies as well (lines 541-543, 547-558).
Also, we referred to the recommended literatures.
For the English proofreading, we asked for a native English professional.
We hope our revisioning meets your expectations.
Reviewer 2 Report
The article deals with a very interesting and up-to-date topic, but there are several drawbacks that should be corrected before publication:
- The introduction provides sufficient background and shows the authors' acquaintance with the works of the field; however, the following changes should be implemented:
- The authors should clearly present the novelty of the paper and how it contributes to the existing knowledge.
- The research problem and aim should be stated in the introduction part.
- All the graphs should be removed from the introduction.
- The literature review is well-written, howevere there is a lack of recently published articles. I suggest considering the following:
-
- Musa, H., Natorin, V., Musova, Z., & Durana, P. (2020). Comparison of the efficiency measurement of the conventional and Islamic banks. Oeconomia Copernicana, 11(1), 29-58. https://doi.org/10.24136/oc.2020.002
- Skare, M., & Porada-Rochoń, M. (2019). Financial and economic development link in transitional economies: a spectral Granger causality analysis 1991-2017. Oeconomia Copernicana, 10(1), 7-35. https://doi.org/10.24136/oc.2019.001
- Stankova, M., Tsvetkov, T., & Ivanova, L. (2019). Tourist development between security and terrorism: empirical evidence from Europe and the United States. Oeconomia Copernicana, 10(2), 219-237. https://doi.org/10.24136/oc.2019.011
- Özer, M., & Karagöl, V. (2018). Relative effectiveness of monetary and fiscal policies on output growth in Turkey: an ARDL bounds test approach. Equilibrium. Quarterly Journal of Economics and Economic Policy, 13(3), 391-409. https://doi.org/10.24136/eq.2018.019
- Damoska Sekuloska, J. (2018). Causality between foreign direct investment in the automotive sector and export performance of Macedonian economy. Equilibrium. Quarterly Journal of Economics and Economic Policy, 13(3), 427-443. https://doi.org/10.24136/eq.2018.021
3. Methodology is well-structures, however, the authors do not explain how they check the model for validity. I.e. reliability tests should be explained.
4. Research results are well presented and could be left as they are.
Author Response
Response to Reviewer 2 Comments
Point 1: The introduction provides sufficient background and shows the authors' acquaintance with the works of the field; however, the following changes should be implemented:
- The authors should clearly present the novelty of the paper and how it contributes to the existing knowledge.
- The research problem and aim should be stated in the introduction part.
- All the graphs should be removed from the introduction.
Response 1: Agree with the reviewer. We enclosed these issues in lines 97-99, 101, 112, 121-125. We removed the graphs from the introduction part.
Point 2: The literature review is well-written, however there is a lack of recently published articles. I suggest considering the following:
- Musa, H., Natorin, V., Musova, Z., & Durana, P. (2020). Comparison of the efficiency measurement of the conventional and Islamic banks. Oeconomia Copernicana, 11(1), 29-58. https://doi.org/10.24136/oc.2020.002
- Skare, M., & Porada-Rochoń, M. (2019). Financial and economic development link in transitional economies: a spectral Granger causality analysis 1991-2017. Oeconomia Copernicana, 10(1), 7-35. https://doi.org/10.24136/oc.2019.001
- Stankova, M., Tsvetkov, T., & Ivanova, L. (2019). Tourist development between security and terrorism: empirical evidence from Europe and the United States. Oeconomia Copernicana, 10(2), 219-237. https://doi.org/10.24136/oc.2019.011
- Özer, M., & Karagöl, V. (2018). Relative effectiveness of monetary and fiscal policies on output growth in Turkey: an ARDL bounds test approach. Quarterly Journal of Economics and Economic Policy, 13(3), 391-409. https://doi.org/10.24136/eq.2018.019
- Damoska Sekuloska, J. (2018). Causality between foreign direct investment in the automotive sector and export performance of Macedonian economy. Quarterly Journal of Economics and Economic Policy, 13(3), 427-443. https://doi.org/10.24136/eq.2018.021
Response 2: Agree and thank you for the suggested literature. We referred to these papers (with the exception of Stankova et al) in the Literature Review part, in lines 138-140, 206-210, 194-196, and 220-223.
Point 3: Methodology is well-structured; however, the authors do not explain how they check the model for validity. I.e. reliability tests should be explained.
Response 3: Agree. We (extended our model specification in lines 346-356, and) tested for robustness as documented in lines 560-567.
Point 4: Research results are well presented and could be left as they are.
Response 4: Thank you.
For the English proofreading, we asked for a native English professional.
We hope our revisioning meets your expectations.
Reviewer 3 Report
The paper entitled
Islamic financial depth, Financial intermediation, and sustainable Economic Growth: ARDL Approach
is sufficient documented regarding the financial literature
and fine structured.
The Abstract provide enough information about the scope of the paper.
Introduction shows states preliminary details about
the literature regarding the topic of the paper. The Information part
is well documented.
Chapter 2. Literature Review
provides enough documentation regarding the scope of the article. The papers presented
are in line with topic and refers to latest research.
Chapter 3 Data and Research Methods
provides enough information about the methodology and data.
Methodology is clearly explained.
The data is no described. We suggest to insert the data description table and also
to provide some data comparison regarding data used in similar studies.
Chapter 4. Empirical Results and Discussion and Chapter 5.Implication and Limitation provides enough information about the study, results, limitations and future research.
The references used in paper are large/extensive - at around 57 items. The articles cited are in line with the topic and also contains latest research in the field.
Author Response
Response to Reviewer 3 Comments
Point 1: Islamic financial depth, Financial intermediation, and sustainable Economic Growth: ARDL Approach is sufficient documented regarding the financial literature and fine structured.
The Abstract provide enough information about the scope of the paper.
Introduction shows states preliminary details about the literature regarding the topic of the paper. The Information part is well documented.
Chapter 2. Literature Review provides enough documentation regarding the scope of the article. The papers presented are in line with topic and refers to latest research.
Chapter 3 Data and Research Methods provides enough information about the methodology and data.
Methodology is clearly explained.
The data is no described. We suggest to insert the data description table and also to provide some data comparison regarding data used in similar studies.
Chapter 4. Empirical Results and Discussion and Chapter 5.Implication and Limitation provides enough information about the study, results, limitations and future research.
The references used in paper are large/extensive - at around 57 items. The articles cited are in line with the topic and also contains latest research in the field.
Response 1: Thank you for your review. Agree with the reviewer concerning the data description. We enclosed this issue in lines 342-344.
For the English proofreading, we asked for a native English professional.
We hope our revisioning meets your expectations.
Round 2
Reviewer 1 Report
I would suggest acceptance.
Author Response
Thank you for your suggestion for the minor spell check; we asked for English proofreading by a native English. Please see the modifications in the attached file.
